# Visualizing near-coexistence of massless Dirac electrons and ultra-massive saddle point electrons

Abhay Kumar Nayak[1†], Jonathan Reiner[1], Hengxin Tan[1], Huixia Fu[1],

Henry Ling[2], Chandra Shekhar[3], Claudia Felser[3], Tami Pereg-Barnea[4],

Binghai Yan[1], Haim Beidenkopf[1†], Nurit Avraham[1†],

[1] Department of Condensed Matter Physics, Weizmann Institute of Science, Rehovot, Israel.

[2] 208-5800 Cooney Road, Richmond, British Columbia V6X3A8, Canada  [3] Max Planck Institute for Chemical Physics of Solids, D-01187 Dresden, Germany.

[4] Department of Physics, McGill University, Montréal, Québec H3A 2T8, Canada.

[†] Corresponding authors: abhaykumar.nayak@weizmann.ac.il, nurit.avraham@weizmann.ac.il, haim.beidenkopf@weizmann.ac.il

**Strong singularities in the electronic density of states amplify correlation effects and play a key role in determining the ordering instabilities in various materials. Recently high order van Hove singularities (VHSs) with diverging power-law scaling have been classified in single-band electron models. We show that the 110 surface of Bismuth exhibits high order VHS with unusually high exponent of $-0.7$. Detailed mapping of the surface band structure using scanning tunneling microscopy and spectroscopy shows that this singularity occurs in close proximity with Dirac bands located at the center of the surface Brillouin zone. The enhanced power-law divergence is shown to originate from the anisotropic flattening of the Dirac band just above the Dirac node. Such near-coexistence of massless Dirac electrons and ultra-massive saddle points enables to study the interplay of high order VHS and Dirac fermions.**

# introduction

The energy dispersion of the Bloch states govern key physical properties of a given material while also regulating the density of states (DOS) which plays a pivotal role in ascertaining the ground state of interacting particles. Several exotic quantum phases, such as, superconductivity [1, 2], charge density wave [3, 4], and magnetism [5, 6] may manifest due to upsurge in the DOS. In two-dimensions, the vanishing electron velocity at a saddle point in the energy dispersion gives rise to logarithmic divergence in the DOS [7], called van Hove singularity (VHS). These VHSs are usually accompanied by a topological transition (Lifshits transitions) in which the Fermi surface transforms between a hole-like and an electron-like form due to the appearance or collapsing of a pocket in the Fermi surface. When the Fermi level lies in the vicinity of such Van Hove points, electron-electron interactions are enhanced, promoting the formation of electronic instabilities and correlated states. In addition to ordinary VHS, two-dimensional systems can also host high-order VHS which is characterized by a power-law divergence of DOS in energy. Such high order singularities display more exotic Fermi surface topological transitions around which the dispersion is flatter than near a conventional van Hove point. These stronger divergences in DOS can further enhance the formation of complex quantum phases via interactions and can play important roles in transport phenomena[8, 9].

High order VHS have been proposed to exist in several electronic systems such as twisted bilayer graphene and trilayer graphene, as well as other Van der Waals compounds, where the band structure can be tuned by changing the twist angle, pressure, or interlayer bias voltage [8, 10–14]. They have been associated with the unusual Landau level structure in biased bilayer graphene [10], the nontrivial thermodynamic and transport properties in $Sr_3Ru_2O_7$ [15]x, the so called supermetal with diverging susceptibilities in the absence of long-range order [16] and more recently with the electronic symmetry breaking in $CsV_3Sb_5$ [17]. The effect of a high order VHS on electron interaction has been studied theoretically in [8, 9, 16] in a system with weak electron interactions. Further theoretical studies classified high-order critical points based on topology, scaling, and symmetry, showing that high order VHS can be realized at generic or symmetric momenta by tuning a few parameters such as twist angle, strain, pressure, and/or external fields[18, 19]. It has been shown that they can be obtained by tuning the parameters in the Hamiltonian, particularly at high

symmetry points in the Brillouin zone. Nevertheless, the corresponding symmetries also restrict the type of singularities that can be achieved.

Bismuth (Bi) is an ideal material to explore the interplay of topological and correlated phases owing to its high spin-orbit coupling and long mean free path [20–23]. Although Bi has been studied in various contexts, such as unconventional superconductivity [24], electron fractionalization [25] and quantum hall ferromagnetism [26, 27], its exact topological classification has been resolved only recently, as being on the verge of a phase transition between a high order topological insulator and a strong topological insulator [28, 29]. Here, we show that the (110) surface of Bi exhibits a remarkable power-law divergence of the DOS arising from high-order VHS presented on this surface. Spectroscopic mapping of the surface band structure at the corresponding energy range, using quasi particle interference (QPI) measurements, shows that this high-order VHS occurs in conjunction with a highly anisotropic Dirac band located at the center of the surface Brillouin zone. By directly visualizing the Dirac bands in QPI we observe a rapid anisotropic flattening of the upper Dirac band just above the Dirac node which leads to the sharp increase in the DOS. Calculated QPI using Green's function approach exhibit excellent agreement with our QPI measurements. The near-coexistence of massless Dirac fermions and heavy saddle-point fermions is an interesting property of the (110) surface that enables to study the interplay and competing orders of high orders VHS and Dirac fermions.

# Results

Fresh surfaces of Bi were exposed by cleaving GdPtBi crystals grown in Bi flux [28, 30]. Due to its anisotropic rhombohedral structure cleaving these crystals under ultrahigh vacuum conditions exposes surfaces of residual crystalline Bi inclusions in various orientations. Here, we measured the Bi(110) and Bi(111) surfaces (Fig.1a), at 4.2K in a low temperature Unisoku STM. A representative topography of pristine Bi(110) surface terraces is shown in Fig.1b and a height profile across a mono-atomic step edge in Fig.1c. A typical differential conductance ($dI/dV$) profile measured over the Bi(110) surface is shown in Fig.1d. The spectrum was measured on a clean terrace far from any impurities. Peaks in the $dI/dV$ spectrum (marked by dashed lines Fig.1d) signify increased local density of states (DOS). Many of them correspond to band extrema shown in the

density functional theory (DFT) calculation of the Bi(110) surface band structure (Fig.1e). Owing to the inverted bulk band structure of bismuth [21, 28], Dirac bands are realized on its various surfaces. While on the Bi(111) surface the Dirac bands are directly overlapping with bulk bands [20, 21, 28], on the (110) surface the Dirac bands at $\Gamma$ and $M$ (Fig.1e) are very well separated from the bulk projected bands, making it ideal to directly visualize their exact dispersion.

## Visualizing a topological Dirac cone on Bi(110)

To resolve the topological Dirac bands and other surface bands on the Bi(110) surface we map the QPI patterns imprinted in the local DOS due to scattering off surface step edges [31]. A spatially resolved spectroscopic $dI/dV$ map measured across an atomic step-edge is shown in Fig.2a (see also Fig.S1 and S2). The momentum along the step edge is conserved and we only see scattering processes with momentum transfer in the direction perpendicular to the step edge. Strong spatial modulations emanating from the step edge and dispersing in energy are embedded in the local DOS. Fourier decomposition of these QPI patterns (Fig.2b) separates them according to the transferred momentum, $\mathbf{q}$, between the scattered electronic states [31]. We identify two prominent scattering processes labeled and marked by yellow dashed lines in Fig.2b. Particularly interesting is the outer QPI pattern (process 1) that exhibits clear linear dispersion over more than a hundred meV, at energies just below the Dirac point at $\Gamma$. This process provides a direct visualization of the topological Dirac band as it results from scattering between the lower part of the Dirac cone to its upper part that curves downwards in a Rashba like manner, as shown in Fig.2c. A similar larger energy window measured on different step-edge (Fig.2d) reproduces the scattering processes marked in Fig.2b along with other dispersing modes at lower energies (see also S3, S4 and S5). To fully associate the observed QPI patterns with particular scattering processes, we compare the data with Green's function calculation of the QPI (Fig.2e; see also Fig.S6), based on *ab initio* calculation of the Bi(110) surface states. The four identified processes (yellow dashed lines) exhibit good agreement in $\mathbf{q}$ values as a function of energy with the corresponding calculated QPI patterns.

Further identification of the particular processes is obtained by comparing the momentum transfer with particular $\mathbf{q}$ vectors that connect the corresponding electron and hole pockets in the calculated surface band structure. A few representative contours of constant energy (CCE) cuts along

with the identified scattering wave vectors (labeled and marked by green arrows) is shown in Fig.2f (see also Fig.S3 and Fig.S4 for details). Sparsity of the surface bands in the relevant energies allowed us to readily visualize and uniquely identify the various scattering processes. The scattering process 1 along the $\Gamma - X_1$ direction, indeed corresponds to momentum transfer between the lower part of the Dirac band, the circular dispersing band around gamma, and its upper part that curves downwards (Fig.2c), giving rise to the U-shaped dispersing pocket. Process 2 results from scattering across the Brillouin zone, between the high intensity tips of those U-shaped pockets. Finally, processes 3 and 4 result from scattering between high intensity points along the $\Gamma - X_1$ direction within the Brillouin zone.

## High-order van Hove singularity

We now focus on the energy region just above the Dirac node in which the upper part of the Dirac band flattens out as a function of energy. The $dI/dV$ spectrum in that energy window exhibits a remarkable increase characterized by a sharp peak at $E_0 = 246$ meV (Fig.1d). This VHS is characterized by a power law divergence with an unusually high exponent of $b = -0.70(2)$. Much higher than an exponent of 0.25 measured near high-order VHS in magic angel graphene [8]. Comparison with DFT calculations shows that the peak at $E_0 = 246$ meV is very well captured by the calculated DOS stemming from the vicinity of the Dirac cone at $\Gamma$ (Fig.3a and Fig.S8 for more details). A power-law fit ($dI/dV \sim (E - E_0)^b$) to the $dI/dV$ profile shown in Fig.3a, for energies above the peak, is presented in Fig.3b along with a log-log plot of the exact fitting region (inset). The power-law exponent was extracted by fitting over the linear region shown in the log-log plot and its corresponding domain in the raw $dI/dV$ profile (dark blue circles in Fig.3b). This remarkable divergence has been observed across all our measurements done at different regions on the Bi(110) surface (see Fig.S9). A similar analysis of the power-law divergence of the DOS for energies below the peak is shown in Fig.3c. The corresponding power-law fit yields a much lower exponent of $b = -0.2$, possibly due to the DOS contributed by other bands. The high order VHS observed on the (110) surface can be contrasted with a conventional VHS observed on the (111) surface, at about 180 meV, as demonstrated by the $dI/dV$ spectrum of the Bi(111) surface shown in Fig.4a and b. Here the VHS originates from six ordinary saddle points that form around $\Gamma$ at

180 meV (Fig.4c) when the six hole pockets along the $\Gamma - M$ direction merge with the hexagonal electron pocket around $\Gamma$ (see Fig.S7). Indeed, the divergence exhibits logarithmic behaviour as expected for conventional VHS (Fig.4b).

Close inspection of the Bi(110) band structure around $\Gamma$, in the corresponding energy window, reveals two Lifshitz transitions at slightly different energies, $\sim 198$ meV and $\sim 229$ meV, which give rise to two pairs of saddle points located along the $\Gamma - X_1$ and the $\Gamma - X_2$ directions, respectively (Fig.S10b and e). While the points along $\Gamma - X_1$ are clearly ordinary saddle points, the nearly tangential band touching along the $\Gamma - X_2$ direction (Fig.3d and e) have similar shape to the $C_2$ symmetric VHS observed on twisted bilayer graphene [8] with a slight deviation just around the touching point. Indeed, probing the DOS on a smaller region just around these nearly tangential touching points does not account for the large exponent we observe. However, on a larger scale (Fig.3a) the power-law exponent extracted on the Bi(110) surface is significantly higher than the one observed on twisted bilayer graphene [8]. We therefore believe that the enhanced power-law divergence does not originate from the immediate local surrounding of the saddle points but rather from the unique topology of the Bi(110) band structure, at this energy range, on a larger scale. As shown in Fig.3e and Fig.S10, the surface bands along the $\Gamma - X_1$ direction that constitute the nearly tangential saddle points remain flat almost across the whole Brillouin zone (Fig.3f). At higher energies these flat bands merge with the hole pockets along $\Gamma - X_2$, such that around the peak energy $\sim 246$ meV, this surface band becomes flat along both the $\Gamma - X_1$ and the $\Gamma - X_2$ directions (Fig.3e and S10), though not to the same extent. This extended flattening seems to give rise to the sharp increase in the DOS we observe in our measurements. While point singularities have been theoretically explored and classified [8, 19], singularities of lines or circles of critical points have not been classified and explored theoretically [19], further studies are needed to generalize the classification to include singularities of infinite dimensions. The bulk chemical potential in Bi(110) lies around 200 mev above the touching point of the Dirac point and the higher order saddle point. Nevertheless, our spectroscopic study may very well motivate thin film growth and investigation [32, 33] of this exotic orientation under efficient gate tuning of the electronic density towards the extreme van Hove singularities and adjacent Dirac node.

# Summary

In summary, we observe a power-law divergent DOS stemming from high-order VHS on the (110) surface of Bismuth. This high order VHS does not originate from the immediate local vicinity of the saddle points hosted on the surface, but rather from an extended flattening of the surface band structure on a larger scale. We further show that this high-order singularity occurs in close proximity to the surface Dirac cone of Bi(110) surface. The coexistence of Dirac fermions and nearly flat bands have been studied theoretically on 2D systems with square-octagon lattice [34]. It was shown that the competition of paramagnetic contribution of the high-order VHS and diamagnetic contribution of the Dirac cones may lead to a dia- to paramagnetic phase transition in orbital suseptibility as a function of doping at the touching point at which the high-order VHS and Dirac point are present. Our observation renders Bi(110) an interesting system to study physical quantities related to high-order VHS in general and to this unique coexistence in particular. Moreover, tuning of the chemical potential towards the high-order VHS in Bi(110) thin films may induce correlated states of matter within strongly interacting topological bands.

# Figures

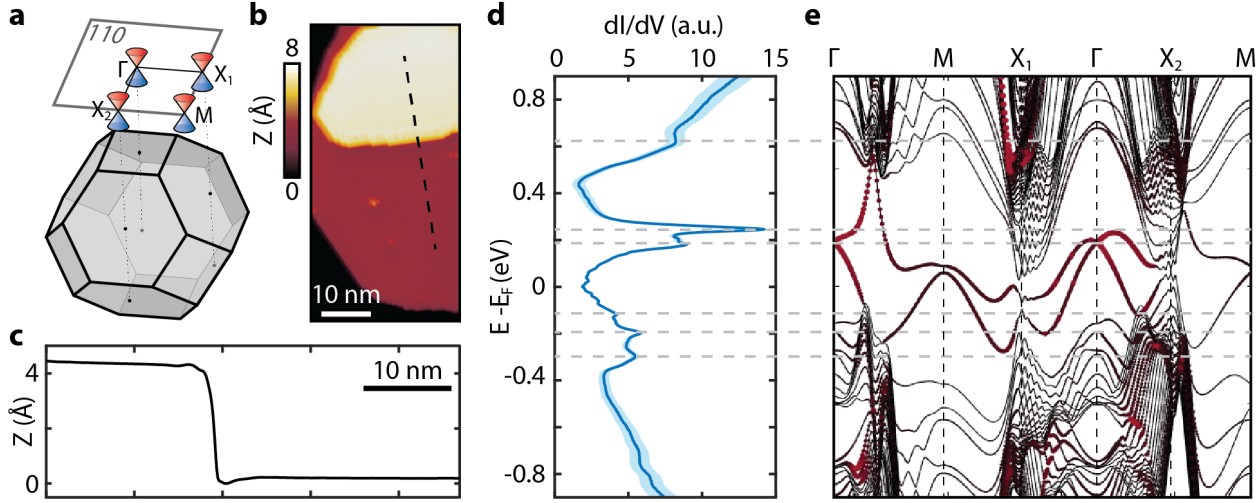

**Figure 1. Electronic structure of Bi(110). a,** Schematic illustration of the bulk Brillouin Zone and the projected 2D Brillouin Zone on the 110 surface with Dirac cones at the TRIM points. **b,** Topography of the Bi(110) surface. **c,** Height profile of an atomic step edge along the dashed line marked in **b**. **d,** An average spectrum ($dI/dV$) measured on the pristine Bi(110) surface (solid blue). The shaded region marks the variation within 95% confidence interval in the $dI/dV$ profile. **e,** Calculated band structure of the Bi(110) surface along the high symmetry lines. Additionally, small gaps of surface bands at $\Gamma$ and $M$ are artificially caused by the finite size effect of the slab model and should vanish in reality.

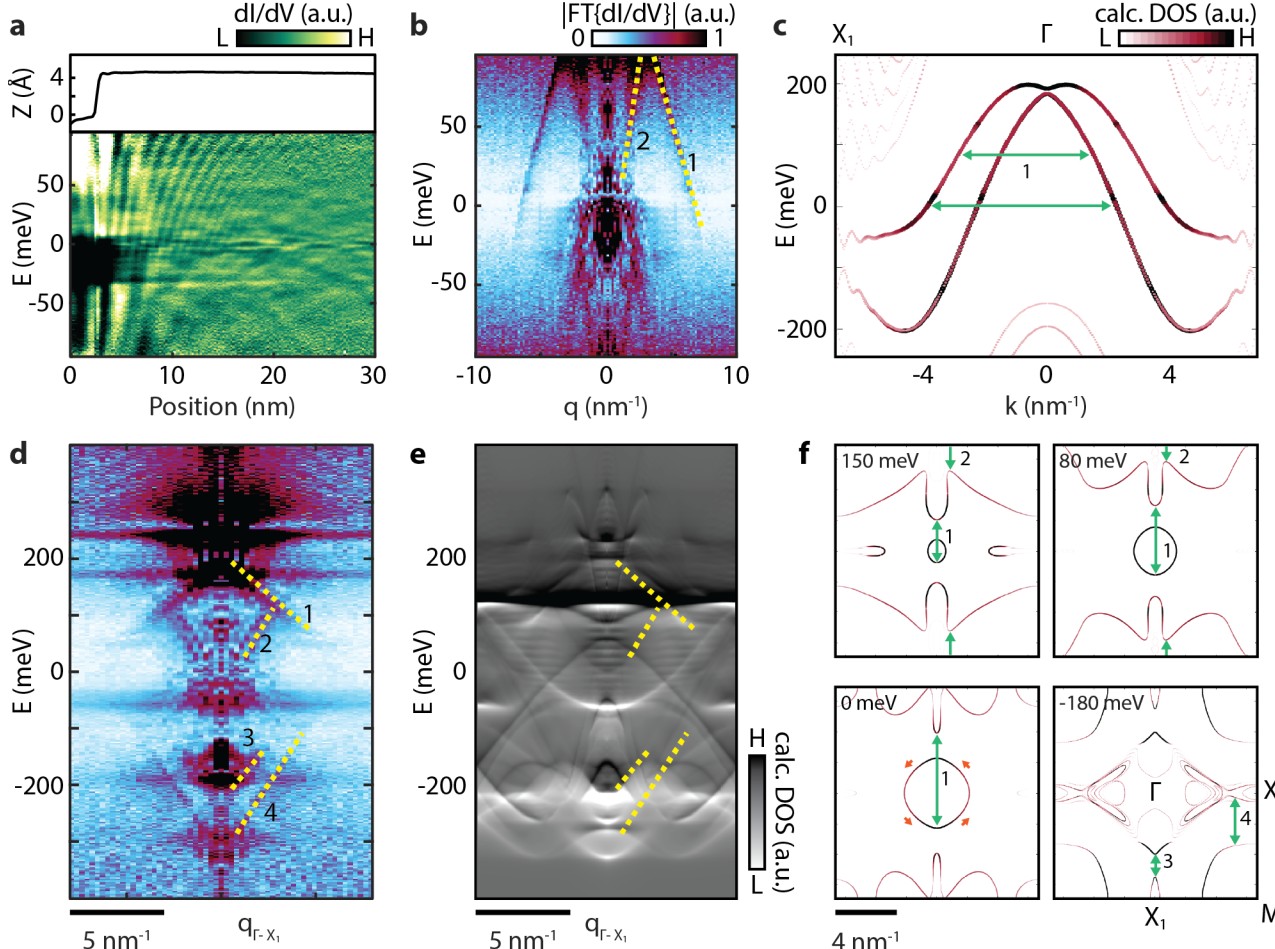

**Figure 2. Quasi-particle interference of a Dirac cone. a,** Topographic profile along a line perpendicular to the step edge and the corresponding $dI/dV$ measurement along the line are shown in the upper and the lower panel, respectively. **b,** Fourier transform (FT) of **a** along the position axis showing the energy dispersion of the scattering wave vectors (marked by the dashed green lines) along $\Gamma - X_1$. **c** Ab initio calculation of the band structure along $\Gamma - X_1$. **d,** Fourier transform of a large energy window $dI/dV$ measurement (similar to **b**) showing the various dispersing modes. **e,** Calculated QPI along $\Gamma - X_1$ using the Green's function approach. The different modes are marked by dashed green lines in **c** and **d**. **f,** Ab initio calculation of contours of constant energy (CCE) at different energies for the Bi(110) surface. The relevant scattering wave vectors are are marked by the green arrows.

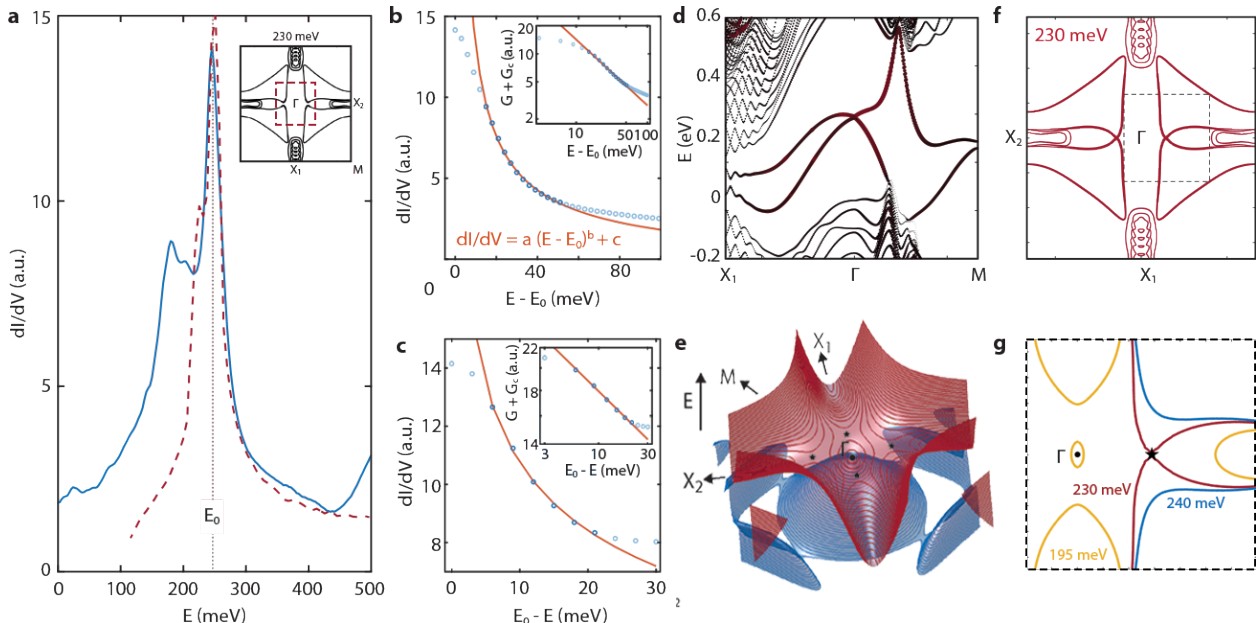

**Figure 3. Power-law divergence in DOS on Bi(110).** **a,** Zoomed in $dI/dV$ profile around the $E_0$ = 246 meV peak overlaid with the calculated density of states (DOS) originating around Γ as marked by the dashed box in the inset. **b,** $dI/dV$ profile to the right of the peak at $E_0$ extracted from **a**. A power-law fit (red) to the raw data (dark blue circle) yields an exponent of b = -0.7. Inset shows a linear fit (red) to the log-log $dI/dV$ profile, indicating a power-law divergence in DOS due to high-order van Hove singularity. **c,** Same as in **b** for the $dI/dV$ profile to the left of the peak at $E_0$, yielding a power-law exponent of b=-0.2. **d,** Calculated band structure of the Bi(110) surface along a high symmetry line showing the highly anisotropic Dirac bands. **e,** Calculated surface band structure of Bi(110) in the vicinity of the Lifshitz transition. **f,** The surface band structure at 230 meV. **g,** The zoomed in surface band structure, of the dotted square area in f, in orange, red and black correspond to 195 meV, 230 meV and 240 meV, respectively.

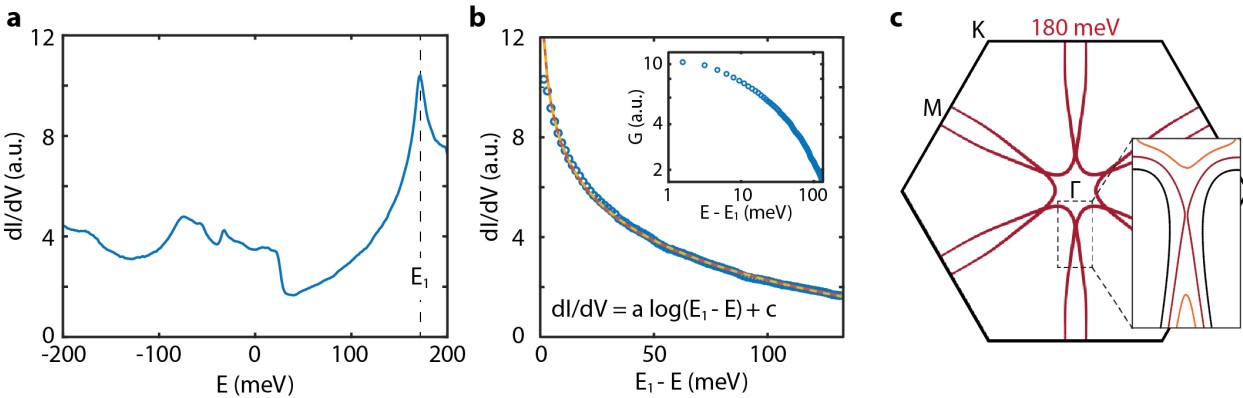

**Figure 4. Logarithmic divergence in DOS on Bi(111). a,** $dI/dV$ profile measured on a pristine Bi(111) surface showing the characteristic peak at $E_1$. **b,** Zoomed in $dI/dV$ profile to the left of $E_1$ in **a** and a corresponding logarithmic fit (orange). Inset shows the corresponding log-log plot. **c,** Calculated band structure of Bi(111) surface in the vicinity of the Lifshitz transition at about 180 meV ($E_1$). Inset: The zoomed in surface band structure in orange, red and black correspond to 100 meV, 180 meV and 260 meV, respectively.

# Methods

## STM measurements

The measurements were performed in a commercial Unisoku STM at 4.2 K. The Pt-Ir tips were characterized in a freshly prepared Cu(111) single crystal. This process ensured a robust tip with reproducible results across different cleaves and samples from different batches. All the $dI/dV$ measurements were taken using standard lock-in techniques.

## DFT calculations

To study the electronic properties of Bi(110) surface, *ab initio* calculations based on density functional theory (DFT) were performed using Vienna Ab initio Simulation Package (VASP)[35, 36]. The projector-augmented wave pseudopotential and a plane-wave energy cutoff of 120 eV were adopted. A 58 Å thick slab was constructed to model the cleaved Bi(110) surface. The in-plane lattice parameters of 4.25 ×4.72 Å and a vacuum region of $\geq$ 15 Å along z-direction were set for the slab model. A $12 \times 12 \times 1$ k-mesh was adopted for sampling the two-dimensional Brillouin Zone. The four upmost atomic layers were relaxed until the force on each atom is less than 0.01 eV/Å. The surface-atom-projected band structures in Fig.1e present the surface electronic properties. The size of red circules represent the projected weight of the top four Bi layers. Because of the finite size effect, the top and bottom surface states have weak hybridization and generate tiny gaps at $\Gamma$ and $M$ points. Such gaps have marginal effects in our analysis of VHS and should vanish in reality.

# Data availability

The data that support the plots within this paper and other findings of this study are available from the corresponding authors upon reasonable request.

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

See Supplemental Material at [URL will be inserted by publisher] containing sections S1-S7 and figures S1-S10.

# Acknowledgement

The authors thank A. Stern and R. Quieroz for insightful discussions. N.A., H.B., and B.Y acknowledge the German–Israeli Foundation for Scientific Research and Development (GIF grant no. I-1364-303.7/2016). B.Y. acknowledges financial support by the Willner Family Leadership Institute for the Weizmann Institute of Science, the Benoziyo Endowment Fund for the Advancement of Science, the Ruth and Herman Albert Scholars Program for New Scientists, and the Israel Science Foundation (ISF 1251/19). T.P. acknowledges financial support from NSERC and FRQNT.

# Author contribution

A.K.N. and J.R. acquired the data. A.K.N analyzed the data. A.K.N, N.A and H.B. conceived the experiments. H.T., H.F. and B.Y. calculated the ab initio model. H.L., and T.P. calculated the theoretical model. C.S. and C.F. grew the material. A.K.N., N.A., and H.B. wrote the manuscript with substantial contributions from all authors.

# Competing interests

The authors declare no competing interests.

# Supplementary Information for

# Visualizing a Dirac cone in proximity to high-order van Hove singularities

Abhay Kumar Nayak[1†], Jonathan Reiner[1], Hengxin Tan[1], Huixia Fu[1],

Henry Ling[2], Chandra Shekhar[3], Claudia Felser[3], Tami Pereg-Barnea[4],

Binghai Yan[1], Haim Beidenkopf[1†], Nurit Avraham[1†],

[1] Department of Condensed Matter Physics, Weizmann Institute of Science, Rehovot, Israel.

[2]208-5800 Cooney Road, Richmond, British Columbia V6X3A8, Canada [3] Max Planck Institute for
Chemical Physics of Solids, D-01187 Dresden, Germany.

[4] Department of Physics, McGill University, Montréal, Québec H3A 2T8, Canada.

[†] Corresponding authors: abhaykumar.nayak@weizmann.ac.il, nurit.avraham@weizmann.ac.il,
haim.beidenkopf@weizmann.ac.il

## S1    Additional QPI measurements

Similar QPI patterns were observed in the dI/dV measurements on two different samples as shown
in Fig.S1 and S2, respectively. The dI/dV profile measured far from any defects and impurities
on the (110) surface is shown in Fig.S1a. The measured profile shows the characteristic peak at
$E \sim 240$ meV. The dI/dV map measured perpendicular to a step edge (marked by black arrow)
is shown in Fig.S1b. It has dispersing features emanating from the step edge and non-dispersing
features associated with the surface band extrema and van Hove singularities. Fourier transform
of the dI/dV map shows the evolution of several dispersing features. The magnitude and the phase

of the Fourier transform is shown in Fig.S1c and d, respectively. Most of the features are well identified in the calculated QPI. The phase of the QPI pattern may reveal information related to the Berry curvature of the surface bands [1, 2]. However, here we use the phase as a means to clearly visualize the scattering processes. Similar QPI features can be seen in another measurement on a different sample, as shown in S2.

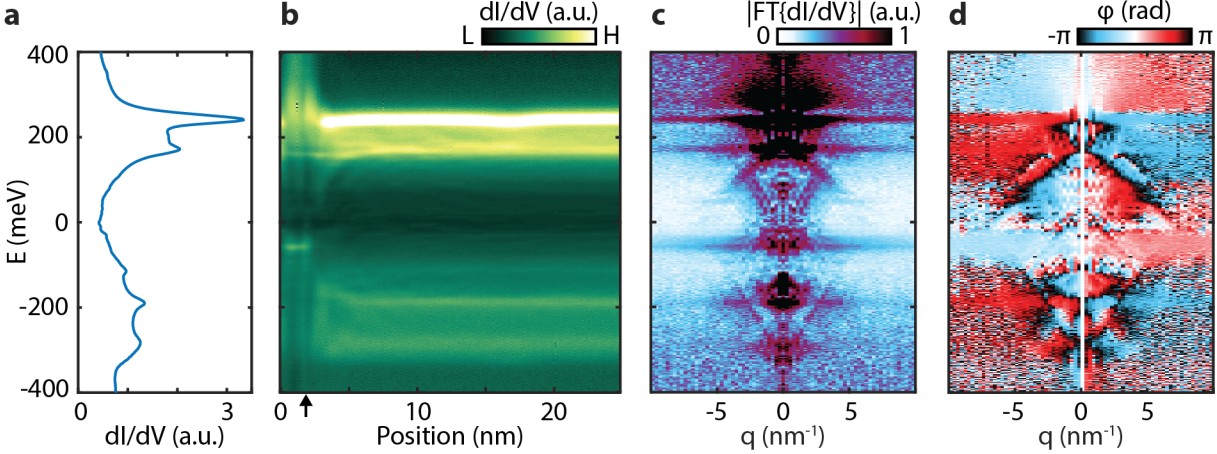

**Fig. S1.** *dI/dV* **map on Bi(110) sample #1. a,** *dI/dV* profile measured far from the step edge. **b,** *dI/dV* mapping of the QPI patterns due to scattering off the step edge (marked by black arrow). **c,d,** The magnitude (same as Fig.3c) and the phase ($\phi$) of the Fourier transform of **b** showing the various dispersing modes, respectively.

## S2   Identification of scattering processes

The detailed scattering processes giving rise to the observed QPI patterns is presented in Fig.S3 and Fig.S4. Fig.S3a shows a high energy resolution QPI map of Bi(110) surface. We clearly identify the main scattering process observed in the QPI map as the intra-Dirac scattering along $X_1 - \Gamma - X_1$, as shown in Fig.S3b and c. Such scattering processes are not forbidden since it occurs across the same Dirac band with a finite overlap in their spin texture (Fig.S5). Similarly, more scattering process are identified in the large energy window QPI map, as shown in Fig.S4.

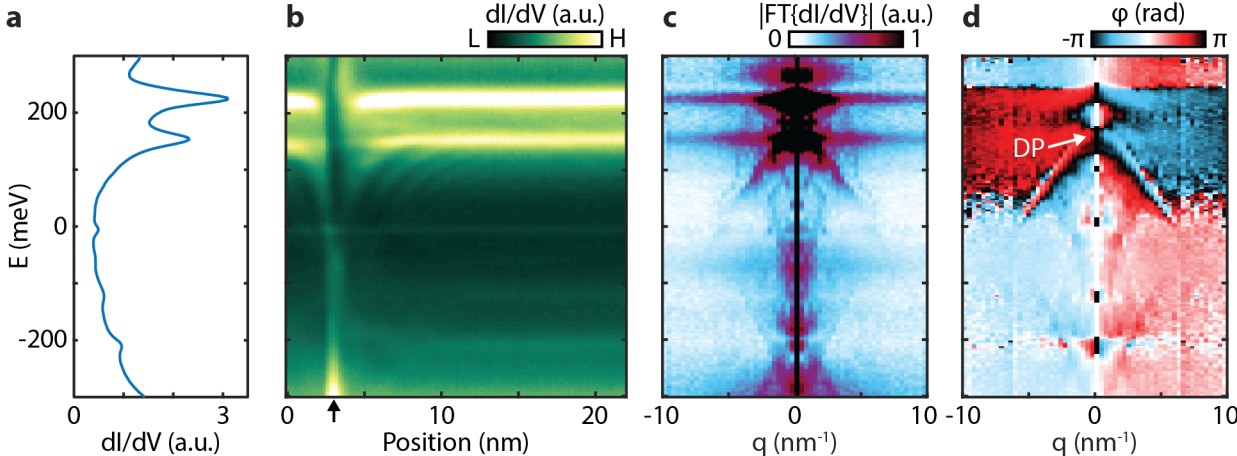

**Fig. S2.** *dI/dV* **map on Bi(110) sample #2.** Similar measurement as Fig.S1 for sample #2.

## S3   Calculated QPI pattern

In order to calculate the QPI pattern we use the Born approximation which states that

$$\delta N(q,\omega) = -\frac{1}{\pi}V(q)\text{Im}\left(\int \frac{d^d k}{(2\pi)^d}G(k,\omega)G(k-q,\omega)\right). \tag{1}$$

In other words, the spatial modulations in the density of states at energy $\omega$, $\delta N(r,\omega)$ is a consequence of scattering events due to the potential $V(r)$. When Fourier transformed, the local density of states is arranged according to the momentum transfer $q$ and hence it is a convolution of the Green's function with itself. This convolution relates on-shell states with energy $\omega$ (where the Green's functions are peaked) such that the initial and final momenta are separated by the vector $q$.

In order to construct the Green's functions we use the wavefunctions which are found by DFT to have the largest weight on the surface and write the retarded Green's function as:

$$G(k,\omega) = \sum_n \frac{|\Psi_{k,n}\rangle\langle\Psi_{k,n}|}{\omega - \varepsilon_{k,n} + i\eta} \tag{2}$$

where $n$ goes over all relevant eigen modes, $\varepsilon_{k,n}$ is its corresponding energy and $\eta$ is a small phenomenological broadening of the states. $\Psi_{k,n}$ is a spinor whose direction is found by DFT.

## S4   DOS of Bi(111)

The DOS measured on a prisitne Bi(111) surface is shown in Fig.S7a (more details in ref.[3]). It shows a characteristic peak at $E_1 \sim 180$ meV. This peak corresponds to the band extrema along

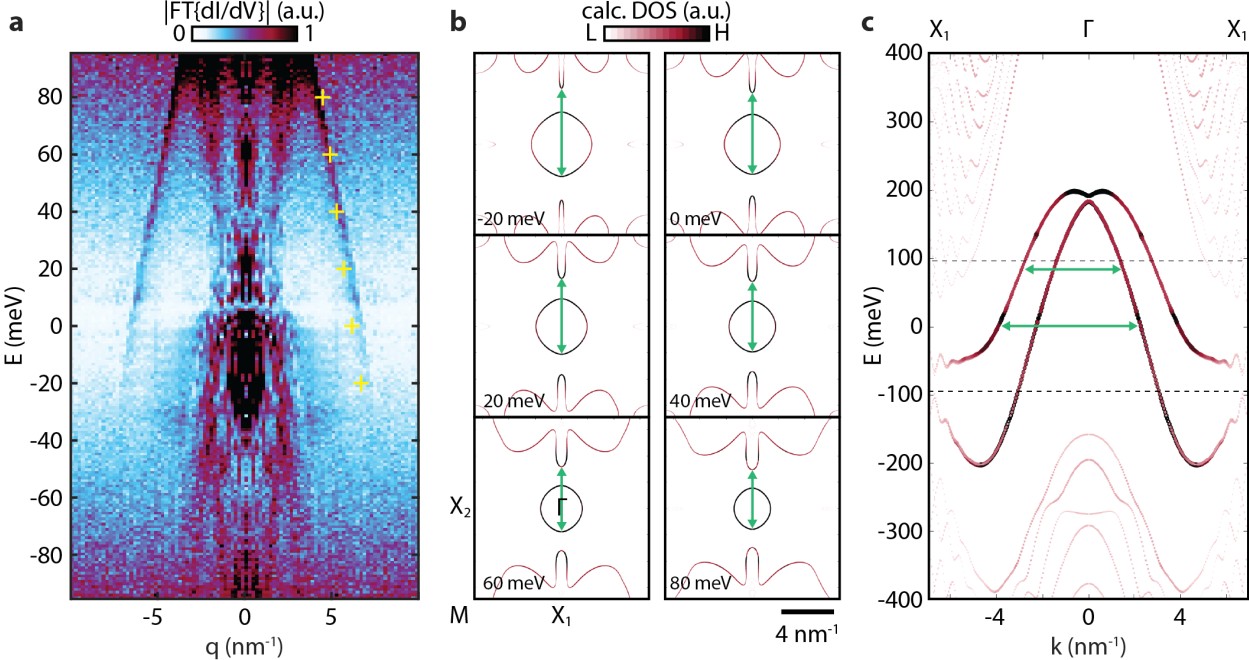

**Fig. S3. Scattering across Dirac bands. a,** Fourier transform of the measured $dI/dV$ profile on Bi(110) surface (same as Fig.1b). **b,** Calculated CCEs for the Bi(110) surface. The scattering process at various energies is marked by the green arrows. The corresponding **q** values are plotted in **a** as yellow '+'. **c,** Calculated band structure of Bi(110) surface along the high-symmetry direction $X_1 - \Gamma - X_1$.

the $\Gamma - M$ direction as shown in Fig.S7b. The DOS diverges logarithmically close to the peak $E_1$ due to ordinary van-Hove singularities (Fig.S7c-e). The topology of the Fermi surface undergoes a transition at the peak energy. This demonstrates the ability to distinguish ordinary and high-order van Hove singularities in dI/dV profiles.

# S5   Calculated DOS on Bi(110)

We extracted the total DOS from Density Functional Theory (DFT) calculated surface band structure. The calculated (calc.) DOS was integrated only over a region close to the $\Gamma$ point as shown in the inset of Fig.S8a. This region (marked by a blue box) includes the high-order van-Hove singularity close to the $\Gamma$ point. The calculated DOS to the right of the peak $E_0$ fits well to a power-law model as shown in Fig.S8b. The same data was plotted in log-log to unambiguously determine its

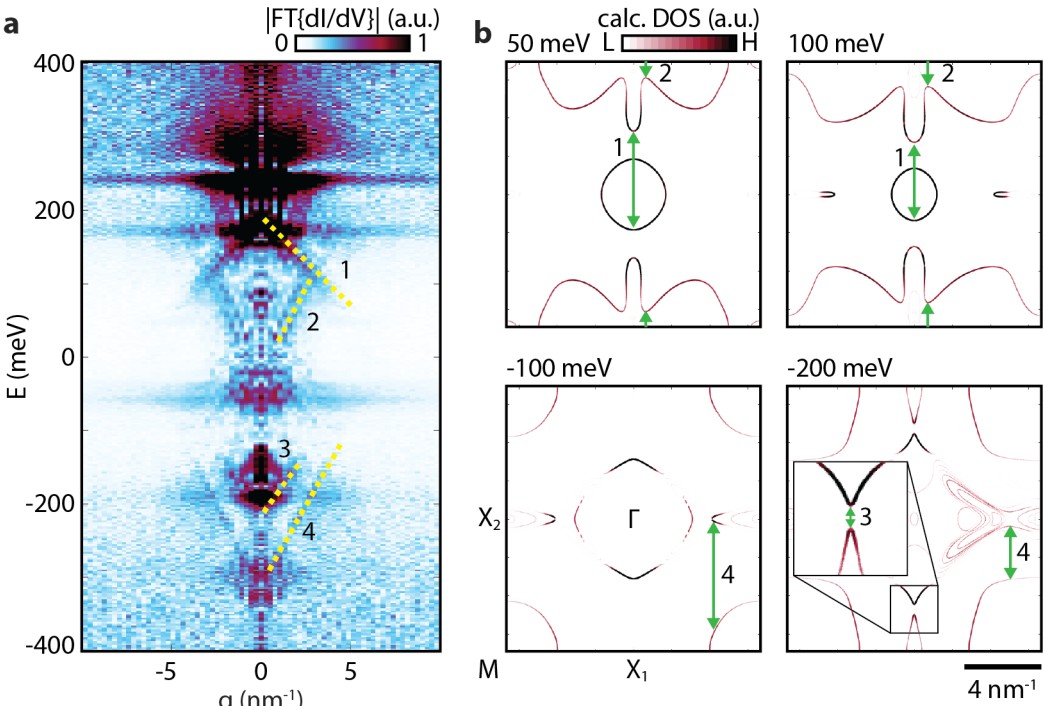

**Fig. S4. Various scattering wave vectors. a,** Fourier transform of dI/dV measurement (same as Fig.3c). **b,** Scattering wave vectors are labeled and marked (green arrows) on some of the representative CCEs of the Bi(110) surface band structure.

power-law nature (inset Fig.Fig.S8b).

# S6   Power-law divergent DOS

The power-law divergence in the density of states was carefully measured across measurements on different regions of the sample as shown in Fig.2 and S9. The spatially averaged dI/dV profile measured on the Bi(110) surface is shown in Fig.S9a (lower panel). The dI/dV profile was averaged over a clean region on the (110) terrace (upper panel Fig.S9a). The dI/dV profile shows a characteristic peak, marked by $E_0$, corresponding to van Hove singularity in the surface band structure. The *ab initio* calculated total DOS (black dashed line), overlaid on the dI/dV profile, shows good agreement (see Fig.S8 for details). We examined the power-law behavior in the density of states in the shaded region of Fig.S9a.

First, we plot the dI/dV profile in log-log to clearly identify the linear regime (marked by dark

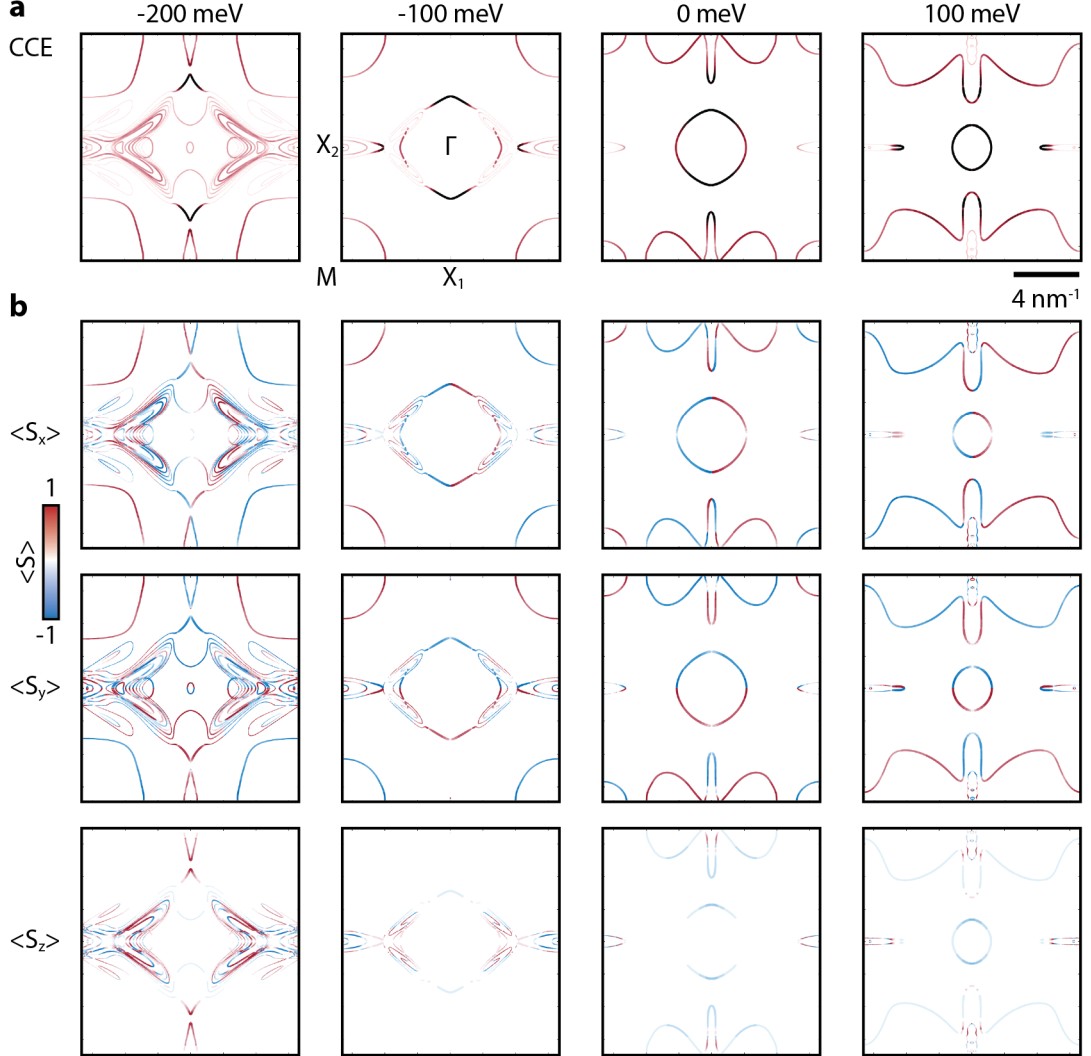

**Fig. S5. Calculated spin texture of the Bi(110) surface. a,** CCEs of the Bi(110) band structure at representative energies. **b,** Normalized spin components of the corresponding CCEs in **a**.

blue circles in Fig.S9b). We fit this subset of the dI/dV profile (that is linear in log-log plot) to a power-law formula $a(E - E_0)^b + d$. This fit yields the following parameters: $a = 14(6)$, $b = -0.7(2)$, and $d = -0.06(30)$. To accurately extract the slope from the log-log plot, we add the constant $d$ extracted from the power-law fit to the DOS in the log-log plot. Therefore, $G + G_C$ is equivalent to $\log(dI/dV - d)$ in Fig.S9b. Fitting a linear model, $m(E - E_0) + c$, to the log-log plot over the relevant energy window (marked by dark blue circles) yields the slope $m = 0.7(2)$, consistent with the power-law fit. The dI/dV profile close to the peak ($|E - E_0| \leq \sim 10$ meV) deviates from the power-law behavior, possibly due to instrumental broadening of the divergence.

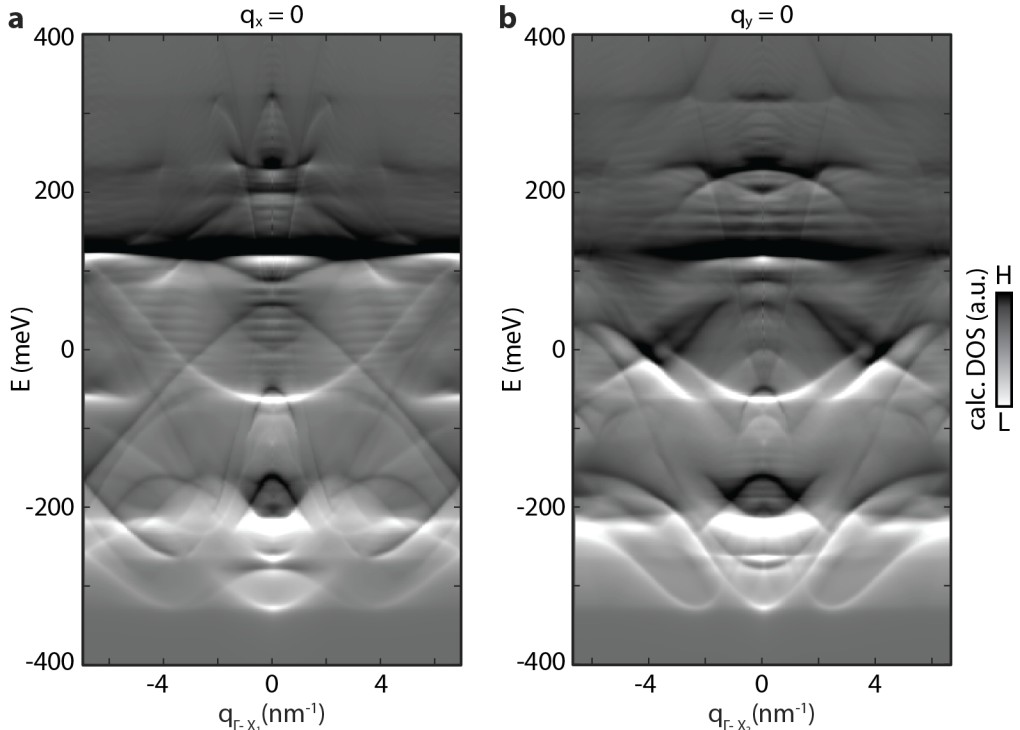

**Fig. S6. Calculated QPI using Green's function approach. a,b,** Calculated QPI cut along $\Gamma - X_1$ and $\Gamma - X_2$ direction, respectively.

## S7 CCEs of Bi(110)

The contours of constant energy (CCE) calculated for the Bi(110) surface is shown in Fig.S10. The CCEs show the transition in the Fermi pockets known as Lifshtiz transition. The first saddle point appears along $\Gamma - X_1$ at $E = 198$ meV and the second saddle point appears along $\Gamma - X_2$ at $E = 229$ meV. The Dirac cone at $\Gamma$ merges with the pockets along $\Gamma - X_1$ (Fig.S10b) creating long straight segments in the CCE as shown in Fig.S10d. Subsequently, the long straight segments merge with the pockets along $\Gamma - X_2$ almost tangentially as shown in Fig.S10e.

## Supplementary References

1. Dutreix, C., González-Herrero, H., Brihuega, I., Katsnelson, M. I., Chapelier, C. & Renard, V. T. Measuring the Berry phase of graphene from wavefront dislocations in Friedel oscillations. *Nature,* 1–4 (2019).

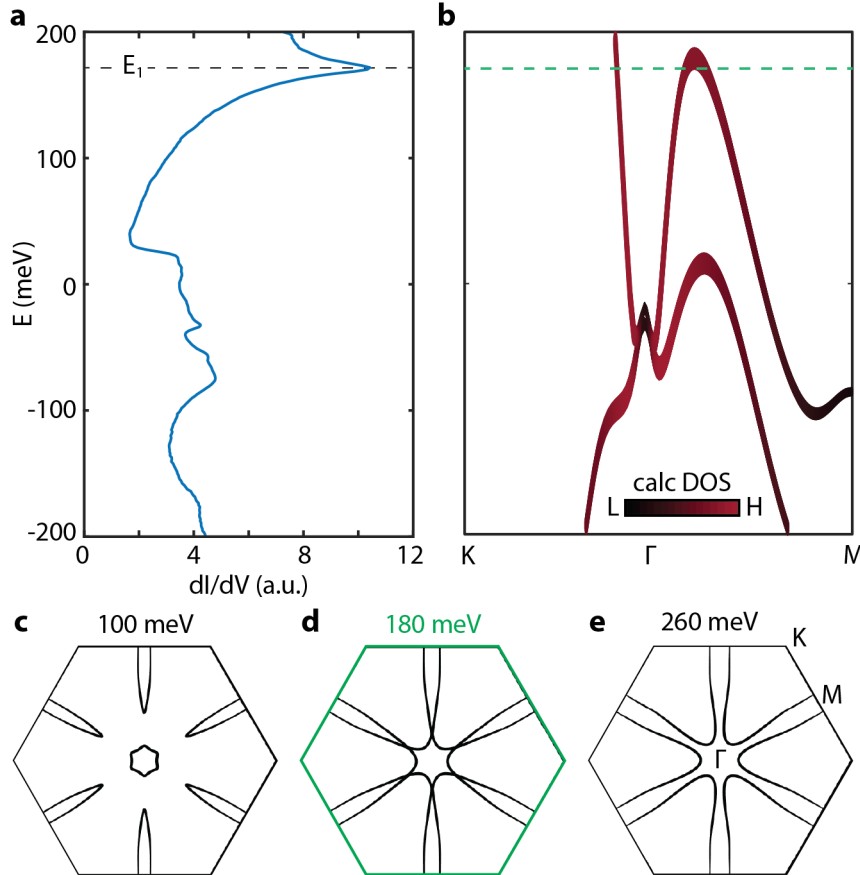

**Fig. S7. Calculated band structure of Bi(111) surface. a,** A typical $dI/dV$ profile measured on a clean (111) surface (same as Fig.2f). **b,** Ab initio calculated band structure of the Bi(111) surface along a high symmetry line. **c-e,** Contours of constant energy showing the evolution of the bands in the vicinity of the Lifshitz transition (green) responsible for the $E_1$ peak.

2. Zhang, Y., Su, Y. & He, L. Local Berry Phase Signatures of Bilayer Graphene in Intervalley Quantum Interference. *Phys. Rev. Lett.* **125,** 116804 (2020).

3. Nayak, A. K., Reiner, J., Queiroz, R., Fu, H., Shekhar, C., Yan, B., Felser, C., Avraham, N. & Beidenkopf, H. Resolving the topological classification of bismuth with topological defects. *Sci. Adv.* **5,** eaax6996 (2019).

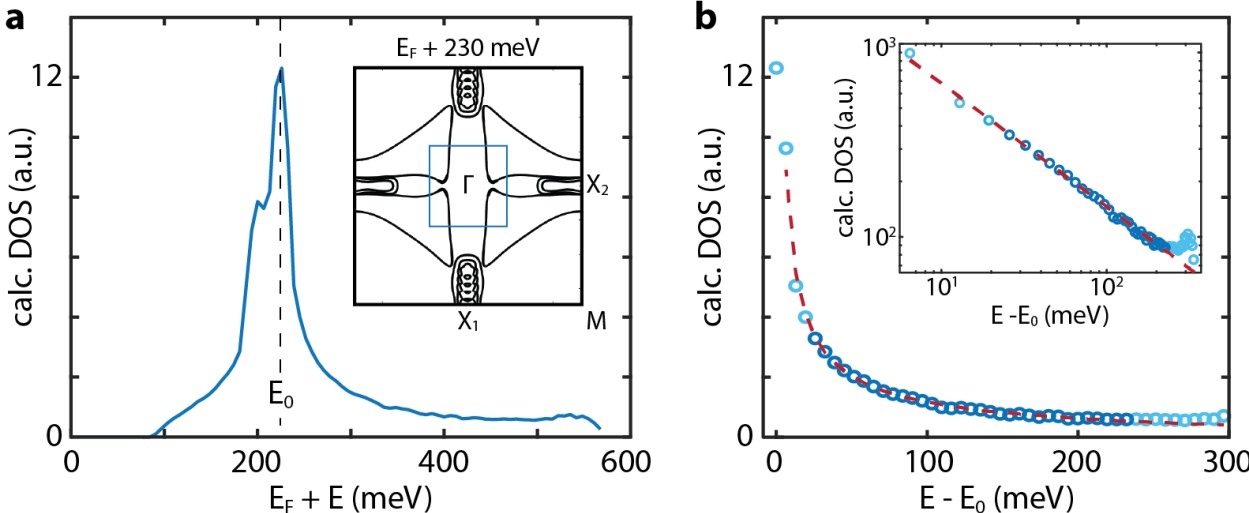

**Fig. S8. DFT calculated DOS on Bi(110). a,** Momentum integrated DOS extracted from DFT surface band structure in the vicinity of the zone center (marked by blue box in inset). Inset: Contour of constant energy at 230 meV above the Fermi energy. **b,** Power-law fit to the calculated DOS, $a(E-E_0)^b + c$. The fit yields an exponent of $b = 0.67(3)$. Inset shows linear fit to log-log plot, $m(E-E_0) + c$. The fit yields a slope of $m = 0.67(2)$.

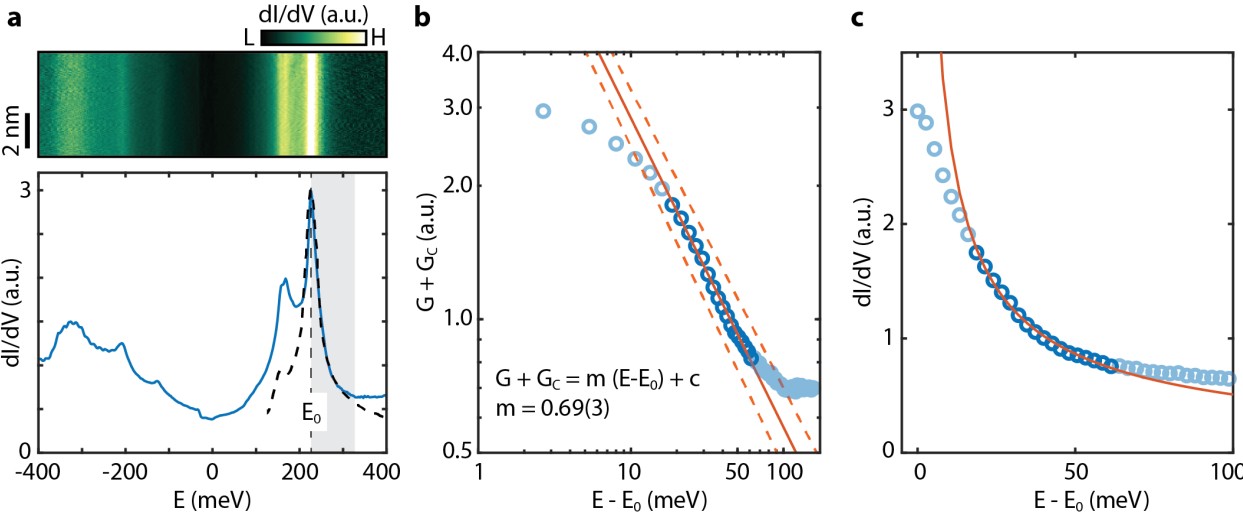

**Fig. S9. Diverging DOS on the Bi(110) surface. a,** dI/dV map on Bi(110) away from defects (upper panel) and spatially averaged dI/dV profile overlaid with calculated DOS (lower panel). **b-c,** Log-log and linear plot of the dI/dV profile close to the diverging peak ($E_0$) marked by the shaded region in **a**. Linear and corresponding power-law fit to the density of states is shown in **b** and **c**, respectively.

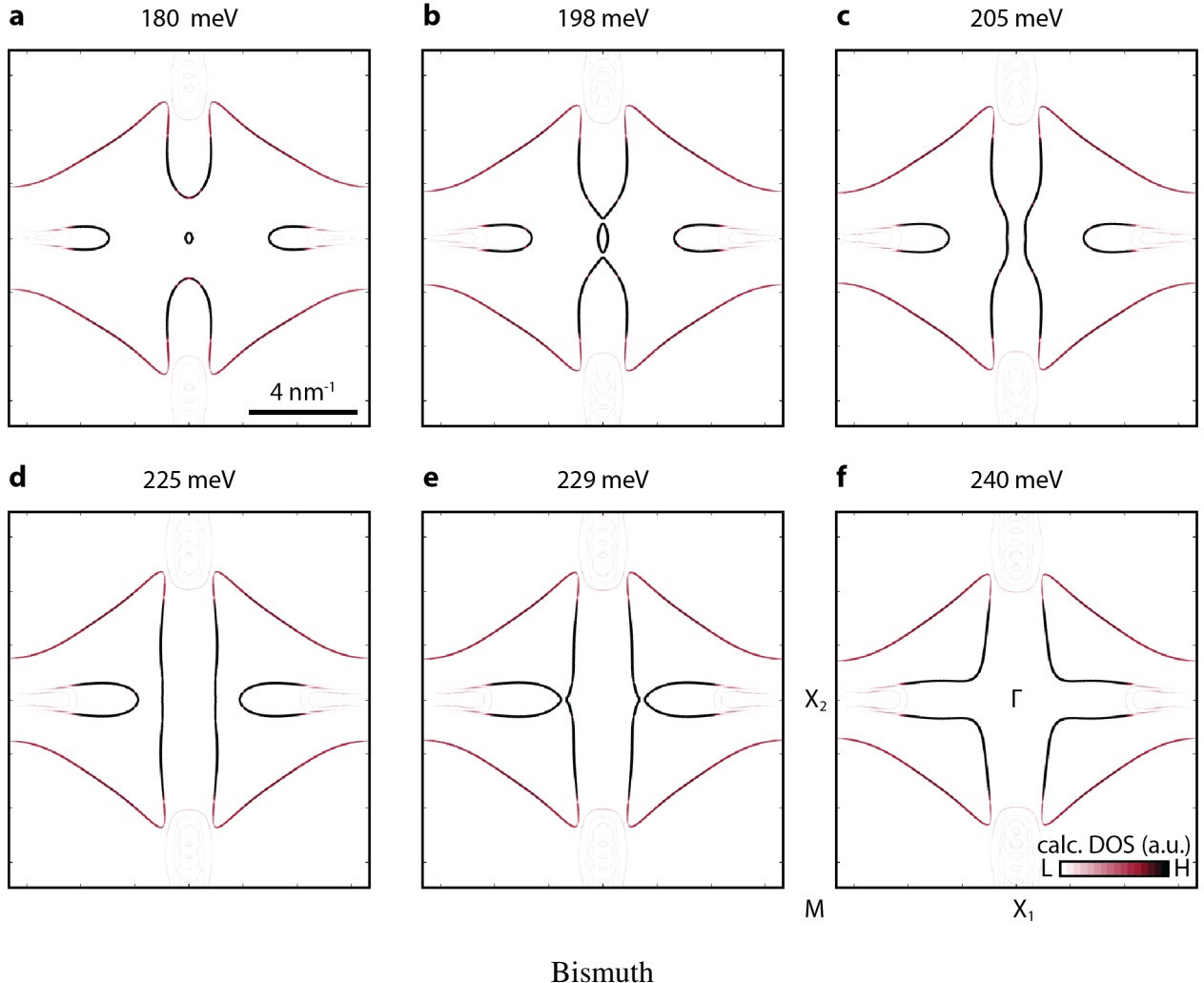

Bismuth

**Fig. S10. Calculated band structure of Bi(110) surface.** Evolution of the Bi(110) surface band structure across Lifshitz transition in **b** and **e**. The nearly tangential band touching in **e** may give rise to high-order van Hove singularity.