# Peer review of "Visualizing near-coexistence of massless Dirac electrons and ultra-massive saddle point electrons"

_SciPost Physics_

## Round 1 · Referee Report · Anonymous (Referee 1) · 2023-7-3

Strengths

(1) A good introduction to the topic, including the survey of relevant literature.
(2) Good motivation for choosing the Bi(110) surface over the (111): The 110 has the surface and bulk bands well separated in energy allowing one to directly probe the former using QPI.
(3) The QPI are directly compared with Green’s function calculations based on ab initio calculations of the Bi(110) surface states to identify microscopic processes contribution to the observed signals.

Weaknesses

(1) The authors motivate the study of higher order van Hove singularities by emphasizing how the enhanced density of states around them can lead to interaction-related electron instabilities. However, the authors do not report any experimental observations supporting such instabilities. Can they comment on why this may be? (This is partly addressed in the three concluding sentences of the Summary from which I infer that it is because the Fermi energy is too far from the DOS singularity.) Given that the DOS divergence computed from DFT (Fig.S8 and Fig.S9) matches the experiment reasonably well, it appears that any interaction effects are well described within the DFT mean-field approximation without any additional symmetry breaking. What would be needed to find a system/material with higher order Van Hove singularities that produce interaction induced symmetry breaking?

Report

In this combined theory-experiment work the authors investigate the effect of strong singularities from saddle points on the DOS and resulting electronic instabilities. The study focuses on higher order van Hove singularities on the 110 surface of Bismuth with a strong DOS divergence ~$1/\omega^{0.7}$. The authors use scanning tunneling microscopy and spectroscopy to show this divergence occurs in close energic proximity to a Dirac point near the \Gamma point of the surface Brillouin zone. The strong divergence is argued to originate from the anisotropic flattening of the bands just above the Dirac point. The system allows one to investigate the interplay of higher order van Hove singularities with Dirac fermions.

Overall, the paper is well written with the results presented in a logical and clear fashion. The results are interesting, and the analysis is thorough. I recommend publication in SciPost.

Requested changes

(1) In the abstract, I suggest being clear about which quantity is diverging and exactly how the exponent is defined, for example $DOS(\omega) \propto \omega^{-0.7}$.
(2) In the abstract I would mention the role that theory played in reaching the conclusions about the band structure. For example, I would modify the existing sentence to the following: “Detailed mapping of the surface band structure using scanning tunneling microscopy and spectroscopy combined with first-principles calculations shows that this singularity occurs in close proximity to Dirac bands located at the center of the surface Brillouin zone. “
(3) For the opening sentence of the introduction I suggest “..ascertaining..” $\to$ “…determining..” since the DOS has a causal effect on the interacting aspects of the physics.
(4) Next line, “..upsurge..” $\to$ ”…an upsurge..”. Following line, “..logarithmic…” $\to$ “.. a logarithmic..”, etc. Please check use of articles throughout manuscript.
(5) In the section on “High-order van Hove singularity”, do the authors mean to quote a value of -0.25 (rather than 0.25) for magic angle graphene? (Also, angle is misspelled “angel” there.)

---

## Round 1 · Referee Report · Anonymous (Referee 2) · 2023-7-9

Strengths

  1. Clear focusing on a topic.
  2. Good presentation of experimental results with identification of main features presented in the spectrum.
  3. Comparison with DFT calculations for the purpose of identifying main observed features in QPI.

Weaknesses

  1. Number of typos (High-order van Hove singularity section, line 5 – typo “magic angel”, also in References)
  2. Conclusions about the origin of van Hove singularities are not fully clear.

Report

Authors performed experiment on 110 surface of Bi in which they identified Dirac cones and strong van Hove singularities in DoS placed close to each other. These results are thus presenting a Bi(110) plane as potentially interesting platform to study physics related to high-order van Hove singularities. The paper is suitable for publication in SciPost Physics after the revisions are made.

Requested changes

  1. What is the assumed energy as a function of momentum dependence of high-order van Hove singularity observed and shown in Fig.3. I would suggest Authors to expand discussion of symmetry properties of material that create and protect this van Hove singularity.
  2. On page 18 the link to Overleaf appears. I would suggest Authors to use standardized ways to publish experimental data and codes for analysis such as Zenodo repositories.
  3. The DoS plots presented in Fig.1 and 3 contain a clear peaks related to the main (high-order) van Hove singularity and additional small peak structure that might be related to usual van Hove singularities. In the dispersion shown in Fig.3f, e, g Authors identify positions of van Hove singularities and the approximate shape of constant energy contours thus concluding about high-order type of saddle points. From these results it might be possible to identify which saddle points exactly produce high-order van Hove singularities and why divergence exponents are different on different sides of the peak in DoS. I would suggest Authors to extend the discussion of these features, as it seems that it is already partially present in the data shown.
  4. What is the energy resolutions in DoS (dI/dV plot) presented in Fig.3
  5. I would suggest Authors to correct typos in the text and in the references list (such as in Refs.15, 16, 34 etc).

---

## Editorial Decision

resubmitted